# OpenReview forum: "On Intriguing Layer-Wise Properties of Robust Overfitting in Adversarial Training"
_TMLR — Accepted by TMLR_

### Review · Reviewer_CrXW · 2024-06-07

**Summary Of Contributions:**

The authors study the problem of robust overfitting, that is, the fact that the test's robust accuracy learning curve has a U-shape. More specifically, in the present contribution, the authors focus on how much different parts of neural networks contribute to robust overfitting. It is demonstrated with many careful experiments that the last layers contribute more to the robust overfitting problem.

Based on this observation, authors modify two methods, weight perturbation and learning rate (related to loss scaling), to act differently on the first and last layers. Both techniques perform reasonably well but do not appear to improve over state-of-the-art methods.

**Audience:**

Yes

**Claims And Evidence:**

Yes

**Requested Changes:**

I would like to stress that I am not an expert in adversarial machine learning, so the present review is a detached perspective from the outside of the field. I suggest Action Editor rely more on reviewers more experienced in the matter.

**Unclear parts**

1. Can authors please include a formal definition of robust accuracy? How robust accuracy is measured in training time? Is it done with the same adversarial?
2. It is not entirely clear what constitutes the last layers. As I understand authors consider classification problems, so the network's output should correspond to probabilities or log probabilities of classes. Is this linear/softmax layer included in the last layer?
3. Does the initialization of the network affect the outcomes of the experiments when some layers are frozen?

**Conflicting evidence**

All the experiments with frozen layers suggest that the last layers' regularisation has more effect on robust overfitting (Figure 3a and Figure 4a). These experiments can be considered a special type of intervention/ablation when one deliberately fixes part of the parameters.

However, in Table 4 we see that AWP performs better than RAT-WP. This case also can be considered a type of intervention to AWP: in AWP adversarial is allowed to perturb all weights, whereas in RAT-WP adversarial perturbs only weights in the last two layers. The result of this intervention suggests that it is better to perturb all weights, and later layers have no special significance.

I would like to ask the authors to:
1. Comment on this possible interpretation of presented data
2. Add results with AWP to tables 1, 2, and 3 -- this will allow one to judge whether perturbation of all layers leads to better robustness

**Overall significance**

Given that AWP performs better than RAT-WP, and RAT-WP does not show state-of-the-art results, it is unclear to what extent the observation made by the authors is significant.

I suggest the authors to:
1. Extend the discussion of comparison between RAT-WP and AWP.
2. Provide additional data that shows differences between RAT-WP and AWP. For example, one may suspect that RAT-WP can lead to higher natural accuracy because it enforces less regularisation. Can the authors provide natural accuracy to all methods in Table 4?

**Strengths And Weaknesses:**

**Strengths**

1. Main claims are supported by most of the experiments
2. Experiments are well documented
3. Main ideas are clearly explained

**Weaknesses**

1. Some unclear parts
2. Some experiments provide conflicting evidence
3. The overall significance of the main claim is unclear

---

> ### Author Response · Authors · 2024-07-17
> **Response to Reviewer CrXW**
>
> We thank Reviewer CrXW for your time and efforts in reviewing this work. Below, we address your main concerns.
>
> >**Q1:** [Unclear parts] Can authors please include a formal definition of robust accuracy? How robust accuracy is measured in training time? Is it done with the same adversarial?
>
> **A1:** Robust accuracy is measured by evaluating the model's performance against adversarial examples.
>
> The measurement of robust accuracy involves generating adversarial examples using a specific attack method, such as Projected Gradient Descent (PGD), and then testing the model on these adversarial examples to calculate the proportion of correctly classified instances.
>
> During training, the attack method used to evaluate robust accuracy remains consistent, such as PGD. However, the adversarial perturbations generated by PGD vary at different stages of training because the generated perturbations change as the model parameters evolve.
>
> >**Q2:** [Unclear parts] Is this linear/softmax layer included in the last layer?
>
> **A2:** The layer distribution of the ResNet used in this paper can be summarized as follows:
>
> Input layer → Residual block 1 (layer1) → Residual block 2 (layer2) → Residual block 3 (layer3) → Residual block 4 (layer4) → Output layer (avgpool and fc)
>
> In this paper, we use layer1-4 to refer to the four residual blocks respectively. The linear/softmax layer is not included in the last layer.
>
> We also tried to include the output layer in layer4, which had no impact on the observed layer-wise property of robust overfitting.
>
> >**Q3:** [Unclear parts] Does the initialization of the network affect the outcomes of the experiments when some layers are frozen?
>
> **A3:** To address your concern, we experimented with different initialization methods, including Xavier and Orthogonal initializations. These variations had almost no impact on the outcomes of the experiments shown in Figure 2. We consistently observed that the optimization of the latter layers exhibits a strong correlation with the robust overfitting phenomenon.
>
> >**Q4:** [Conflicting evidence] Whether perturbation of all layers leads to better robustness. Comment on this possible interpretation of presented data.
>
> **A4:** Perturbing all layers, as done in AWP, can indeed lead to improved model robustness in some cases. However, it is worth noting that adversarial robustness and robust overfitting are two different aspects. The focus/contribution of this work is the layer-wise property of robust overfitting. This work is not to pursue state-of-the-art adversarial robustness but to provide a better understanding of the robust overfitting phenomenon. To the best of our knowledge, the underlying mechanisms of robust overfitting remain unclear. In this work, we found that the latter layers present a strong correlation to the robust overfitting phenomenon, and regularizing the optimization of these layers can effectively mitigate robust overfitting. These findings provide novel insights into understanding the robust overfitting phenomenon better.
>
> >**Q5:** [Overall significance] Extend the discussion of comparison between RAT-WP and AWP. Provide additional data that shows differences between RAT-WP and AWP.
>
> **A5:** To mitigate your concern, we provide the comparison between RAT-WP and AWP on the CIFAR-10 dataset under the $L_\infty$ threat model, as shown in the table below. As you analyzed, RAT-WP enforces less regularization compared to AWP, thus achieving higher natural accuracy. For adversarial robustness, it is observed RAT-WP slightly degrades the model's performance in some cases. However, RAT-WP generally maintains comparable adversarial robustness to AWP, which can be attributed to the strong correlation between the latter layers and robust overfitting.
> | Method | Network | Natural(best) | PGD-20(best) | AA(best) | Natural(last) | PGD-20(last) | AA(last) |
> |----------|----------|----------|----------|----------|----------|----------|----------|
> | AT | PreAct ResNet-18 | 81.16 | 52.31 | 47.95 | 84.44 | 44.45 | 42.05 |
> | AWP | PreAct ResNet-18 | 81.11 | 55.39 | 50.12 | 82.00 | 54.73 | 49.85 |
> | RAT-WP | PreAct ResNet-18 | 82.24 | 54.85 | 49.19 | 83.36 | 53.98 | 48.24 |
> | AT | Wide ResNet-34-10 | 85.49 | 55.57 | 52.13 | 86.50 | 47.37 | 46.09 |
> | AWP | Wide ResNet-34-10 | 85.30 | 58.35 | 54.07 | 85.39 | 57.16 | 53.49 |
> | RAT-WP | Wide ResNet-34-10 | 86.08 | 58.92 | 54.46 | 86.23 | 58.23 | 53.98 |
> | TRADES | PreAct ResNet-18 | 82.77 | 52.67 | 49.28 | 82.94 | 49.65 | 46.80 |
> | TRADES-AWP | PreAct ResNet-18 | 82.05 | 55.44 | 51.42 | 82.57 | 54.81 | 50.73 |
> | TRADES-RAT-WP | PreAct ResNet-18 | 82.61 | 55.33 | 51.34 | 83.08 | 54.85 | 51.04 |
>
> Thank you once again for your valuable feedback. Please do not hesitate to reach out if there are any further clarifications needed.

---

> ### Author Response · Authors · 2024-07-22
>
> Dear Reviewer CrXW,
>
> We want to express our appreciation for your valuable feedback, which greatly helped us improve the quality of this paper. If there are any remaining concerns, we would be glad to have further discussions.
>
> Thanks again for your time, and we look forward to hearing from you.
>
> Best regards,
> Authors

---

> ### Author Response · Authors · 2024-07-24
>
> Dear Reviewer CrXW,
>
> We really appreciate your constructive comments that helped us improve this paper. We have made our best effort to address your concerns. We hope our elaborations and the revisions made in the new manuscript could address your concerns. Please let us know if there is more to clarify.
>
> Thanks again for your time, and we look forward to hearing from you.
>
> Best regards,
> Authors

---

> ### Author Response · Authors · 2024-07-26
>
> Dear Reviewer CrXW,
>
> It has been some time since we last heard from you, and we understand that you might be very busy. We just wanted to send a gentle reminder and see if there are any remaining questions or concerns. We sincerely thank you for reviewing our paper and look forward to hearing from you soon.
>
> Best regards,
> Authors

---

> > ### Comment · Reviewer_CrXW · 2024-08-01
> >
> > I would like to apologize for not being active enough during the discussion period and thank the authors for providing clarifications.
> >
> > After getting familiar with all the other reviews and discussions around them, I decided to vote in favor of accepting the article. It is my understanding that the main points of disagreement are: (i) robust overfitting is a simple consequence of (natural) underfitting and is unrelated to the regularisation of later layers, (ii) the later layers have more parameters, so robust overfitting is simply a matter of the number of parameters being affected by regularisation, (iii) the later layers are more important for NN performance in general, so the result by authors is not surprising/interesting enough, (vi) methods of regularisation proposed by authors are not SOTA.
> >
> > I think the authors convincingly addressed (i) and (ii). Point (iii) is highly subjective and goes against TMLR guidelines that only ask to judge the submission based on how well the claims were supported. The same (excluding the subjective part) applies to point (iv).

---

### Review · Reviewer_bT7r · 2024-06-13

**Summary Of Contributions:**

The paper addresses the problem of overfitting in adversarial training of deep neural networks. It is observed that regularizing the final layers of a deep neural network has a larger effect than regularizing the earlier layers. Based on this observation, it is proposed that regularization be applied to the final layers of the network. In experiments, the authors compare different versions of their regularization scheme to unregularized models.

**Audience:**

Yes

**Claims And Evidence:**

No

**Requested Changes:**

- The authors claim that weight decay is a regularization method that mitigates overfitting (Section 1 p1). This is false.
- In Section 3.1, the authors claim Figure 2 shows that "any settings that do not fix the parameters for layer 4 result in a more severe gap between the best accuracy and the accuracy at the last epoch." This is technically accurate, but the authors should acknowledge the more obvious effect is that fixing the Layer 4 parameters decreases performance! This underfitting is what causes there to be no overfitting. The way it is phrased in the paper is misleading and needs to be revised.
- It is misleading to claim this phenomena as "a general property of adversarial learning" (Section 3.1, second paragraph).  The authors need to clarify their claim as being either: (1) this is a property particular to adversarial training, in which case they need to give evidence that it doesn't affect non-adversarial training, or (2) clarify to the reader that this is a general property of deep learning. I believe the latter is true. It is well known that weight updates in the last layer are particularly important in deep learning in general (not just in adversarial training), so for the experiments in Figure 2, I would expect similar patterns with non-adversarial training.
- In the experiments comparing the proposed regularization methods, the authors report the "best" and "final" accuracy of the model. I think the final accuracy is irrelevant because in practice one would use early stopping. The best accuracy is problematic because taking the maximum of the test set loss over training gives an overestimate of the true generalization performance. The correct way to do this experiment is to have a train/valid/test split where early stopping is performed using the validation set and then the test set performance is reported. I think this is critical for fair experiments.
- The experiments show that some regularization helps performance compared to non-regularized adversarial training. This is no surprise. But are there any experiments showing that applying regularization using the proposed RAT method is an improvement over the existing regularization methods mentioned? To be convincing, these experiments need to be done carefully with hyperparameter optimization on a validation set.

**Strengths And Weaknesses:**

Strengths:
- The paper is mostly well-organized and written clearly.

Weaknesses:
- The logical argument for why regularization should be preferentially applied to the later layers is flawed.
- The experiments have many weaknesses.

---

> ### Author Response · Authors · 2024-07-17
> **Response to Reviewer bT7r**
>
> We thank Reviewer bT7r for your time and efforts in reviewing this work. Below, we address your main concerns.
>
> >**Q1:** The authors claim that weight decay is a regularization method that mitigates overfitting (Section 1 p1). This is false.
>
> **A1:** Weight decay, also known as L2 regularization, penalizes large weights by adding a term to the loss function proportional to the sum of the squared weights. This penalization helps to prevent the model from becoming overly complex, thereby reducing the risk of overfitting.
>
> To address your concern, we propose to revise the relevant expressions as follows:
> “Typical regularization practices to mitigate overfitting, such as L1 & L2 regularization (weight decay), data augmentation, etc. are reported to be as inefficient compared to simple early stopping.”
>
> >**Q2:** The authors should acknowledge the more obvious effect is that fixing the Layer 4 parameters decreases performance! This underfitting is what causes there to be no overfitting. The way it is phrased in the paper is misleading and needs to be revised.
>
> **A2:** The parameter-fixing experiments in Figure 2 illustrate the correlation between the optimization of the latter layers and the robust overfitting phenomenon. It is true that fixing the model parameters of the latter layers decreases performance. However, this evidence does not necessarily imply that underfitting is responsible for the relief of robust overfitting. For example, we observe that fixing parameters in other layers also leads to some degree of underfitting, as shown in Figure 2. However, these measures have almost no effect on the robust overfitting phenomenon.
>
> To address your concern and improve the clarity of our manuscript, we will explain in the revision:
> “While fixing the parameters for the latter layers does lead to decreased performance, indicating underfitting, this does not fully explain the relief of robust overfitting. Notably, fixing parameters in other layers also results in underfitting to some extent, but does not mitigate robust overfitting. This suggests that the interaction between layer-specific parameter optimization and robust overfitting is more intricate and highlights the significance of investigating the layer-wise properties of robust overfitting.”
>
>
> >**Q3:** It is misleading to claim this phenomenon as "a general property of adversarial learning".
>
> **A3:** Thank you for your constructive feedback regarding our claim in Section 3.1. The main focus of this work is robust overfitting, a unique phenomenon in adversarial training. That is, after a certain point during adversarial training, the model’s robust test accuracy continues to substantially decrease with further training. However, robust overfitting is typically absent in natural training (non-adversarial training). For example, naturally trained models can often be trained to zero training error, effectively memorizing the training set, seeming without causing any detrimental effects on the generalization performance [1]. Considering that robust overfitting is unique to adversarial training, we limit our findings and conclusions to the scope of adversarial training.
>
> [1] Rice, Leslie, Eric Wong, and Zico Kolter. "Overfitting in adversarially robust deep learning." International conference on machine learning. PMLR, 2020.
>
> >**Q4:** The final accuracy is irrelevant, and the best accuracy is problematic.
>
> **A4:** Robust overfitting refers to the phenomenon where, after a certain point during adversarial training, the model’s robust test accuracy continues to substantially decrease with further training. The final and best accuracies are reported to reflect the degree of robust overfitting. This evaluation method is commonly used in related works [2,3,4]. Specifically, the greater the difference between the final and best accuracies, the more severe the robust overfitting. Regarding the evaluation of best accuracy, we followed the same experimental settings as outlined in [2,3,4], which are the standard settings for evaluating robust overfitting.
>
> [2] Wu, Dongxian, Shu-Tao Xia, and Yisen Wang. "Adversarial weight perturbation helps robust generalization." Advances in Neural Information Processing Systems. 2020.
> [3] Chen, Tianlong, et al. "Robust overfitting may be mitigated by properly learned smoothening." International Conference on Learning Representations. 2020.
> [4] Zhang, Jingfeng, et al. "Geometry-aware Instance-reweighted Adversarial Training." International Conference on Learning Representations. 2020.

---

> ### Author Response · Authors · 2024-07-17
> **Response to Reviewer bT7r**
>
> >**Q5:** Are there any experiments showing that applying regularization using the proposed RAT method is an improvement over the existing regularization methods mentioned?
>
> **A5:** It is worth noting that adversarial robustness and robust overfitting are two different aspects. The focus/contribution of this work is the layer-wise property of robust overfitting. This work is not to pursue state-of-the-art adversarial robustness but to provide a better understanding of the robust overfitting phenomenon. To the best of our knowledge, the underlying mechanisms of robust overfitting remain unclear. In this work, we found that the latter layers present a strong correlation to the robust overfitting phenomenon, and regularizing the optimization of these layers can effectively mitigate robust overfitting. These findings provide novel insights into understanding the robust overfitting phenomenon better.
>
> To address your concern, we provide the comparison between RAT-WP and AWP on the CIFAR-10 dataset under the $L_\infty$ threat model, as shown in the table below. For natural accuracy, RAT-WP enforces less regularization compared to AWP, thus achieving higher natural accuracy. For adversarial robustness, it is observed that RAT-WP slightly degrades the model's performance in some cases. However, RAT-WP generally maintains comparable adversarial robustness to AWP, which can be attributed to the strong correlation between the latter layers and robust overfitting.
>
> | Method | Network | Natural(best) | PGD-20(best) | AA(best) | Natural(last) | PGD-20(last) | AA(last) |
> |----------|----------|----------|----------|----------|----------|----------|----------|
> | AT | PreAct ResNet-18 | 81.16 | 52.31 | 47.95 | 84.44 | 44.45 | 42.05 |
> | AWP | PreAct ResNet-18 | 81.11 | 55.39 | 50.12 | 82.00 | 54.73 | 49.85 |
> | RAT-WP | PreAct ResNet-18 | 82.24 | 54.85 | 49.19 | 83.36 | 53.98 | 48.24 |
> | AT | Wide ResNet-34-10 | 85.49 | 55.57 | 52.13 | 86.50 | 47.37 | 46.09 |
> | AWP | Wide ResNet-34-10 | 85.30 | 58.35 | 54.07 | 85.39 | 57.16 | 53.49 |
> | RAT-WP | Wide ResNet-34-10 | 86.08 | 58.92 | 54.46 | 86.23 | 58.23 | 53.98 |
> | TRADES | PreAct ResNet-18 | 82.77 | 52.67 | 49.28 | 82.94 | 49.65 | 46.80 |
> | TRADES-AWP | PreAct ResNet-18 | 82.05 | 55.44 | 51.42 | 82.57 | 54.81 | 50.73 |
> | TRADES-RAT-WP | PreAct ResNet-18 | 82.61 | 55.33 | 51.34 | 83.08 | 54.85 | 51.04 |
>
> Thank you once again for your valuable feedback. Please do not hesitate to reach out if there are any further clarifications needed.

---

> ### Comment · Reviewer_bT7r · 2024-07-18
>
> A1: I agree with the author's response. Sorry, I had misread the text and gotten confused.
>
> A2: Clearly underfitting (poor performance on the train set) is highly correlated with a reduction in robust overfitting (defined as the difference between best and final robust test performance). This makes sense, and it is on display in Figure 2. But the authors are claiming that there is some additional phenomenon taking place --- that regularizing the last layer in particular reduces robust overfitting with only minimal effect on the train set performance. The authors are not being careful enough in distinguishing these ideas, and I don't think the paper supplies enough evidence to support this second claim. There is no theoretical argument provided for why this should be the case, so a higher level of evidence is needed to support this claim.
>
> To support their hypothesis, the authors claim in Figure 2 that, "fixing parameters in other layers also results in underfitting to some extent, but does not mitigate robust overfitting." This is demonstrably false. For example, in Figure 2a plot 3, fixing layers [2,3] (yellow) simultaneously reduces robust train performance, robust test performance, and robust overfitting. I grant that the yellow curve has more robust overfitting than the red or green curves (where the last layer is fixed), but it also has better training set performance. So there is no evidence here to support the idea that there is something special about fixing the final layer, beyond the general tendency to inhibit fitting.
>
> A3: The authors claim that robust overfitting is unique to adversarial training. What evidence is there for this? I believe that a DNN trained without adversarial training will see the robust test accuracy increase and then decrease with further training. Thus, robust overfitting is not unique to adversarial training, but a general property of DNNs.
>
> Similarly, the effect of regularizing different layers of a DNN will be different. It is well known that the later layers are particularly important for learning, so regularizing the last layers will result in more regularization than regularizing the early layers. Again, this is not unique to adversarial training, but a general property of DNNs.
>
> A4: I see that previous work has examined the phenomenon of robust overfitting by comparing best and final accuracy. I think this is fine for a qualitative description of the phenomenon. However, the authors are claiming to reduce robust overfitting without completely eliminating it, so it is important to have a better metric. Simply using the difference between the best and final robust test performance after a fixed number of training epochs is problematic because there are so many hyperparameters that will affect this. In the end, the important metrics will be the test accuracy and the robust test accuracy of the final selected model.
>
> A5: The authors say that the goal of this manuscript is not to improve performance but to understand the phenomenon of robust overfitting and how it relates to learning in the different layers. My contention is that the latter layers play an important role in overfitting in general, not just in robust overfitting. So if the goal is to understand the effect of regularizing different layers, the experiments should show the effects on non-adversarially trained DNNs, not just in the adversarial setting.

---

> > ### Author Response · Authors · 2024-07-18
> > **Further Response to Reviewer bT7r**
> >
> > Thanks for your reply. We will address your remaining concerns below:
> >
> > >**Q1:** The authors fail to explain how the regularization of the latter layers somehow mitigates robust overfitting. Clearly underfitting is highly correlated with a reduction in robust overfitting.
> >
> > **A1:** The focus/contribution of this work is the layer-wise property of robust overfitting. However, to the best of our knowledge, the underlying mechanisms of robust overfitting remain unclear. Therefore, we cannot definitively explain how the regularization of the latter layers mitigates robust overfitting.
> >
> > For example, while we understand that it makes sense to correlate underfitting with a reduction in robust overfitting, we cannot conclude that it is the underfitting that causes this mitigation. If that were the case, then any underfitting effect introduced during adversarial training should alleviate robust overfitting to some extent. However, as shown in Figure 2, this is not necessarily true.
> >
> > In summary, based on the existing evidence, we indeed cannot provide a clear explanation for robust overfitting, but this does not compromise our layer-wise property of robust overfitting.
> >
> > >**Q2:** The authors are claiming that there is some additional phenomenon taking place --- that regularizing the last layer in particular reduces robust overfitting with only minimal effect on the train set performance.
> >
> > **A2:** It appears there may be some misunderstandings. In our paper, It is stated that fixing model parameters during adversarial training can severely harm the model’s robustness, referring to the robust test accuracy. Then we introduced our method that strives to reduce robust overfitting with only minimal impact on test set performance.
> >
> > >**Q3:** The authors claim that robust overfitting is unique to adversarial training. What evidence is there for this?
> >
> > **A3:** Robust overfitting was initially observed in PGD-AT [1]. It was then formally defined in the work [2], which showed that traditional remedies for overfitting in deep learning offer limited help in addressing robust overfitting in adversarial training. This prompted further efforts to explore robust overfitting.
> >
> > [1] Madry, Aleksander, et al. "Towards Deep Learning Models Resistant to Adversarial Attacks." International Conference on Learning Representations. 2018.
> > [2] Rice, Leslie, Eric Wong, and Zico Kolter. "Overfitting in adversarially robust deep learning." International conference on machine learning. PMLR, 2020.
> >
> > >**Q4:** It is important to have a better metric. Simply using the difference between the best and final robust test performance after a fixed number of training epochs is problematic because there are so many hyperparameters that will affect this.
> >
> > **A4:** We agree that better metrics can provide a more accurate evaluation of robust overfitting. In addition to quantitative experimental results, such as comparing best and final accuracy, we have also provided numerous learning curves of the models in our paper. These learning curves allow us to capture the extent of robust overfitting more intuitively. We hope the learning curves can address your concerns.
> >
> > >**Q5:** My contention is that the latter layers play an important role in overfitting in general, not just in robust overfitting.
> >
> > **A5:** We agree that the effect of regularizing different layers of a DNN will vary, and the layer-wise property may be a general property of DNNs. However, robust overfitting is typically absent in natural training (non-adversarial training). For example, in the CIFAR-10 dataset, the test accuracy of naturally trained models remains stable in the later stages of training and does not exhibit the abnormal performance degradation observed in adversarial training.
> >
> > Hope the explanations above could address your concerns. Please let us know if there is more to clarify.

---

> > > ### Comment · Reviewer_bT7r · 2024-07-18
> > > **Summary of disagreement**
> > >
> > > The authors have not addressed my concerns.  To summarize:
> > >
> > > 1) It appears "robust overfitting" can be viewed as a more sensitive indicator of overfitting because performance is measured on an adversarial test set, but it is not fundamentally different from the general phenomenon of overfitting. In particular, it occurs during standard training in addition to adversarial training. I see nothing in the previous literature, present manuscript, or rebuttal to contradict this. However, the authors are insisting that the phenomenon only occurs with adversarial training.
> > >
> > > 2) Because robust overfitting is more difficult to avoid, we often observe it even when we do not observe "standard" overfitting. It directly follows that eliminating robust overfitting requires more aggressive regularization. I argue that this simple "null hypothesis" explains all the experimental results presented in the manuscript. The authors claim that something more complicated is happening, but the experiments do not directly compare their hypothesis against this simple hypothesis.
> > >
> > > 3) The experiments are not done carefully enough to differentiate between the authors' hypothesis and the null hypothesis above.

---

> > > > ### Author Response · Authors · 2024-07-19
> > > > **Further Response to Summary of Disagreement**
> > > >
> > > > Thanks for your prompt reply! We will address your further questions below.
> > > >
> > > > Robust overfitting is measured on the adversarial test set, while “standard” overfitting is measured on the natural test set. In fact, these two types of overfitting are fundamentally different. For example, the adversarial test set changes continuously during the model’s training process because adversarial perturbations are generated on-the-fly. At different stages of adversarial training, the model’s adversarial test set is constantly evolving. In contrast, the natural test set remains fixed throughout the entire training process.
> > > >
> > > > We understand the hypothesis that unifying robust overfitting and “standard” overfitting can explain the experimental results presented in the manuscript. We greatly appreciate the reviewer’s valuable feedback. However, we do not have evidence to support this hypothesis that unifies robust overfitting and “standard” overfitting. We humbly request the reviewer to provide the relevant evidence supporting this hypothesis.
> > > >
> > > > Hope the explanations above could address your concerns. Please let us know if there is more to clarify.

---

> > > > > ### Comment · Reviewer_bT7r · 2024-07-20
> > > > > **What would make me change my stance**
> > > > >
> > > > > The authors have not addressed my concerns. Here are some things that would make me rethink my stance on this paper:
> > > > > 1. Evidence that robust overfitting does not exist in standard neural network training. That is, train a neural network without adversarial examples, and at the end of each epoch evaluate its performance on adversarial test examples. If the robust test accuracy increased and plateaued without decreasing (overfitting), I would believe the authors' claim that robust overfitting only affected adversarial training.
> > > > > 1. Evidence that robust overfitting is significantly different from standard overfitting. Previous work shows that there can be robust overfitting even if there is not measurable overfitting, but are there experiments showing an increase in test accuracy but a decrease in robust test accuracy? (I expect one could construct such a scenario since NNs can overfit to some parts of the data while underfitting on others, but observing this phenomena would help strengthen the argument that these are different forms of overfitting).
> > > > > 1. Evidence that the proposed regularization method addresses robust overfitting better than other regularization strategies, controlling for the effects on learning.

---

> ### Author Response · Authors · 2024-07-20
> **Further Response to What would make me change my stance**
>
> Thanks for your reply. We will address your remaining concerns below:
>
> >**Q1:** Evidence that robust overfitting does not exist in standard neural network training.
>
> **A1:** We conduct natural training on the CIFAR-10 dataset and evaluate the model’s performance under the PGD-20 attack on the test set. It is observed that the model maintains 0% robust test accuracy throughout the training process, with no instances of robust test accuracy rising and falling.
>
> >**Q2:** Evidence that robust overfitting is significantly different from standard overfitting.
>
> **A2:** PGD-AT exhibits the phenomenon of an increase in natural test accuracy but a decrease in robust test accuracy. As shown in Figure 1, we can see that during adversarial training, the model’s natural test accuracy remains increasing and stable, while the model’s robust test accuracy continues to decline.
>
> >**Q3:** Evidence that the proposed regularization method addresses robust overfitting better than other regularization strategies
>
> **A3:** Work [1] shows that typical regularization practices to mitigate overfitting, such as l1 & l2 regularization (weight decay), data augmentation, etc., are reported to be inefficient compared to simple early stopping in PGD-AT. Our method, however, outperforms PGD-AT’s best robust accuracy by a large margin, demonstrating the effectiveness of our approach.
>
> [1] Rice, Leslie, Eric Wong, and Zico Kolter. "Overfitting in adversarially robust deep learning." International conference on machine learning. PMLR, 2020.
>
> Hope the explanations above could address your concerns. Please let us know if there is more to clarify.

---

> > ### Comment · Reviewer_bT7r · 2024-07-20
> >
> > __Q1:__ If the model has 0% robust test accuracy at the start, then it is not surprising that we don't observe robust overfitting. The attack model should be weakened in order to see how natural training affects robust test accuracy.
> >
> > __Q2:__ I agree that Figure 1 displays a downward trajectory of the robust test accuracy while the natural test accuracy increases a small amount. I understand this to be the model overfitting in the region of data space surrounding the training points (thus making it easier for the network to be attacked), but still learning general patterns (thus increasing the test accuracy) --- the model is overfitting to some parts of the space while still underfit to other parts. Since this occurs in non-adversarial training as well, I am hoping for something more to tease apart the differences, but I'm not sure what that would be.
> >
> > __Q3:__ Early stopping using a validation set is one of the most effective methods to manage overfitting, but in practice one uses early stopping in combination with other regularization methods. It is a mistake to dismiss these other regularization methods as ineffective. The authors claim their regularization method is particularly well-suited for combating robust overfitting --- I believe it is necessary for them to compare to other regularization methods to support their claim.

---

> > > ### Author Response · Authors · 2024-07-21
> > > **Further Response to Reviewer bT7r**
> > >
> > > Thanks for your reply. We will address your remaining concerns below:
> > >
> > > >**Q1:** If the model has 0% robust test accuracy at the start, then it is not surprising that we don't observe robust overfitting. The attack model should be weakened in order to see how natural training affects robust test accuracy.
> > >
> > > **A1:** First, we evaluated the robustness of the model under perturbation budgets of 2/255, 4/255, 6/255, and 8/255, and observed that the model’s robust accuracy is very close to 0 in all cases. Secondly, in the requested experiment, the training and testing attacks are different, which deviates from the definition of robust overfitting. Therefore, regardless of the experimental results, they cannot illustrate the connection between robust overfitting and natural training. Finally, the requested experiment is rarely encountered in practice, and we currently see no need to investigate problems that are either nonexistent or very rare in practice.
> > >
> > > >**Q2:** The model is overfitting to some parts of the space while still underfit to other parts. Since this occurs in non-adversarial training as well, I am hoping for something more to tease apart the differences, but I'm not sure what that would be.
> > >
> > > **A2:** Robust overfitting and “standard” overfitting are different, as we have clearly explained in the previous response: robust overfitting is measured on the adversarial test set, while “standard” overfitting is measured on the natural test set. The adversarial test set changes continuously during the model’s training process because adversarial perturbations are generated on-the-fly. At different stages of adversarial training, the model’s adversarial test set is constantly evolving. In contrast, the natural test set remains fixed throughout the entire training process. Our work focuses on the former, which is robust overfitting in PGD-AT.
> > >
> > > >**Q3:** It is a mistake to dismiss these other regularization methods as ineffective. I believe it is necessary for them to compare to other regularization methods to support their claim.
> > >
> > > **A3:** We apologize for the confusion caused. Here, the term “ineffective” refers to the impact of these regularization methods on the model’s adversarial robustness. To address your concern, we propose to revise the relevant expressions as follows:
> > > “Typical regularization practices to mitigate overfitting, such as L1 & L2 regularization (weight decay), data augmentation, etc., are reported to be as inefficient in adversarial robustness compared to simple early stopping.”
> > >
> > > Actually, AWP has been compared with other regularization methods, as shown in Figure 3(d) in work [1]. AWP demonstrates superiority over other regularization methods. Our method generally maintains comparable adversarial robustness and similar learning curves to AWP, as shown in Table 4 and Figure 4. We hope the evidence can address your concern regarding the comparison to other regularization methods.
> > >
> > > [1] Wu, Dongxian, Shu-Tao Xia, and Yisen Wang. "Adversarial weight perturbation helps robust generalization." Advances in neural information processing systems 33 (2020): 2958-2969.
> > >
> > > Hope the explanations above could address your concerns. Please let us know if there is more to clarify.

---

> ### Author Response · Authors · 2024-07-26
>
> Dear Reviewer bT7r,
>
> It has been some time since we last heard from you, and we understand that you might be very busy. We just wanted to send a gentle reminder and see if there are any remaining questions or concerns. We sincerely thank you for reviewing our paper and look forward to hearing from you soon.
>
> Best regards,
> Authors

---

### Review · Reviewer_JDM1 · 2024-07-15

**Summary Of Contributions:**

This submission’s main contributions are (i) analyzing robust overfitting in neural networks in a layer-wise fashion and identifying that optimization in latter layers has more effect on robust overfitting; (ii) proposed a general prototype (RAT), where additional regularization is imposed on latter layers of a network, to mitigate robust overfitting. The proposed prototype is instantiated with two types of methods: fixed non-decaying learning rate and adversarial weight perturbation (AWP).

**Audience:**

Yes

**Claims And Evidence:**

Yes

**Requested Changes:**

**Critical**:
1. It appears that Table 1 and Table 4 share the same experiment setup. If this is true, then why does the first row (AT) of Table 4 not correspond to the first row of Table 1, but to the first row of Table 3 instead?

2. In Section 3.1, it is mentioned that “robust overfitting is mitigated at the cost of robust accuracy. For example in AT-fix-param-[3,4], if we leave both layer 3 & 4 unoptimized, robust overfitting will practically disappear, but the peak performance is much worse compared to standard AT”. Are there any motivations for selecting Layer 3&4 instead of Layer 4 only to apply regularizations in Section 4? For other types of network architectures, how does the selection of “latter layers” transfer?

3. In Figure 4(b), why are the front layers “Layer 1&2&3” while the latter layers are “Layer 3&4”? I am confused with the definition of former vs latter layers here.

**Nice to have**:
1. How are the semi-supervised learning methods in Section 2.2 relevant to the methods and observations in this paper? Is there any reason for mentioning them in the Related Work section?

2. In Section 4.2, it is mentioned that “learning rate decay in SVHN’s training does not have much connection to the sudden increases in robust test performance or the prevalence of robust overfitting”. Where is this evidence for the SVHN dataset, is it in Figure 5?

3. In Figure 4(c), it is shown that RAT_WP has a negative robust generalization gap throughout most of the training. Can the authors offer some explanations for this unusual phenomenon? What about AWP applied to all layers?

**Strengths And Weaknesses:**

**Strengths**:
1. I am not familiar with the adversarial training literature, so my review should be taken with a grain of salt. But the authors clearly state how their focus differs from existing works on robust generalization, which convinced me that their investigation of layer-wise contribution to robust overfitting is unique.
2. The two proposed solutions to mitigate robust overfitting are straightforward and should be scalable.

**Weakness**:
1. My main concern is that the latter blocks of ResNets contain more parameters than the former blocks. However, the current paper does not provide insights or analysis on whether the latter blocks’ larger effect on robust overfitting is due to the larger number of parameters or something else perhaps more inherent to the latter layers. Therefore, it is hard to conclude the cause of the empirical observations on the layer-wise effects presented in the paper.

2. The effectiveness of the proposed RAT (e.g. RAT-WP) seems limited, when evidence in Table 4 shows that AWP (which is AWP applied to all layers, if I understood correctly), achieves better performance in both test robustness and robust generalization gap. I understand that in comparison to standard AT, the experiments suggest that regularizations on latter layers have a larger contribution to the reduction of the robust generalization gap. However, again, without an analysis on whether this is simply due to the larger number of parameters in the latter layers, it is hard to see how interesting the findings are.

3. There are no quantitative results of RAT applied with other adversarial training methods like TRADES and with other networks architectures. The authors show visualizations of robust test performance of AWP under TRADES instead of standard AT in Figure 10 and performance of AWP for different network architectures in Figure 9, but it would be good to see some quantitative comparisons like in Table 1 through Table 4. That will help verify the effectiveness of RAT. It might also help to answer my second question in Requested Changes, please see below.

---

> ### Author Response · Authors · 2024-07-17
> **Response to Reviewer JDM1**
>
> We thank Reviewer JDM1 for your time and efforts in reviewing this work. Below, we address your main concerns.
>
> >**Q1:** My main concern is that the latter blocks of ResNets contain more parameters than the former blocks.
>
> **A1:** The latter blocks of ResNets do contain more parameters than the front blocks. However, the layer-wise property of robust overfitting is not actually due to the larger number of parameters in the latter layers. We use proof by contradiction to illustrate this point. Specifically, let's assume that the layer-wise property of robust overfitting is due to the larger number of parameters in the latter layers. Then, in Figure 2, when we fix the parameters of the front layers, the degree of robust overfitting should be alleviated to some extent because the front layers also contain a certain number of parameters. However, when we fix the parameters of the front layers, the degree of robust overfitting remains almost unchanged, as shown in Figure 2. Therefore, the layer-wise property of robust overfitting is not due to the larger number of parameters in the latter layers.
>
> >**Q2:** The effectiveness of the proposed RAT-WP seems limited.
>
> **A2:** Perturbing all layers, as done in AWP, can indeed lead to improved model robustness in some cases. However, it is worth noting that adversarial robustness and robust overfitting are two different aspects. The focus/contribution of this work is the layer-wise property of robust overfitting. This work is not to pursue state-of-the-art adversarial robustness but to provide a better understanding of the robust overfitting phenomenon. To the best of our knowledge, the underlying mechanisms of robust overfitting remain unclear. In this work, we found that the latter layers present a strong correlation to the robust overfitting phenomenon, and regularizing the optimization of these layers can effectively mitigate robust overfitting. These findings provide novel insights into understanding the robust overfitting phenomenon better.
>
> >**Q3:** There are no quantitative results of RAT applied with other adversarial training methods like TRADES and with other network architectures.
>
> **A3:** We provide the quantitative results of RAT-WP and AWP on the CIFAR-10 dataset under the $L_\infty$ threat model with different adversarial training methods and network architectures, as shown in the table below. For natural accuracy, RAT-WP enforces less regularization compared to AWP, thus achieving higher natural accuracy. For adversarial robustness, it is observed that RAT-WP slightly degrades the model's performance in some cases. However, RAT-WP generally maintains comparable adversarial robustness to AWP, which can be attributed to the strong correlation between the latter layers and robust overfitting.
> | Method | Network | Natural(best) | PGD-20(best) | AA(best) | Natural(last) | PGD-20(last) | AA(last) |
> |----------|----------|----------|----------|----------|----------|----------|----------|
> | AT | PreAct ResNet-18 | 81.16 | 52.31 | 47.95 | 84.44 | 44.45 | 42.05 |
> | AWP | PreAct ResNet-18 | 81.11 | 55.39 | 50.12 | 82.00 | 54.73 | 49.85 |
> | RAT-WP | PreAct ResNet-18 | 82.24 | 54.85 | 49.19 | 83.36 | 53.98 | 48.24 |
> | AT | Wide ResNet-34-10 | 85.49 | 55.57 | 52.13 | 86.50 | 47.37 | 46.09 |
> | AWP | Wide ResNet-34-10 | 85.30 | 58.35 | 54.07 | 85.39 | 57.16 | 53.49 |
> | RAT-WP | Wide ResNet-34-10 | 86.08 | 58.92 | 54.46 | 86.23 | 58.23 | 53.98 |
> | TRADES | PreAct ResNet-18 | 82.77 | 52.67 | 49.28 | 82.94 | 49.65 | 46.80 |
> | TRADES-AWP | PreAct ResNet-18 | 82.05 | 55.44 | 51.42 | 82.57 | 54.81 | 50.73 |
> | TRADES-RAT-WP | PreAct ResNet-18 | 82.61 | 55.33 | 51.34 | 83.08 | 54.85 | 51.04 |
>
> >**Q4:** Why does the first row of Table 4 not correspond to the first row of Table 1, but to the first row of Table 3 instead?
>
> **A4:** Thank you for pointing out the inconsistency between Table 1 and Table 4. You are correct that Table 1 and Table 4 share the same experimental setup, and the first row of Table 4 should correspond to the first row of Table 1. Due to an oversight on our part, the data for the first row of Table 4 was incorrectly recorded and instead corresponds to the first row of Table 3.
>
> We will re-check our data and correct the first row of Table 4 in the revision. We appreciate you pointing out this error and apologize for any confusion it may have caused.

---

> ### Author Response · Authors · 2024-07-17
> **Response to Reviewer JDM1**
>
> >**Q5:** Are there any motivations for selecting Layer 3&4 instead of Layer 4 only to apply regularizations in Section 4? For other types of network architectures, how does the selection of “latter layers” transfer?
>
> **A5:** In our experiments, the selection of layers was comprehensively considered based on the model's adversarial robustness and robust overfitting. We conducted ablation experiments with TRADES on the CIFAR-10 dataset using PreAct ResNet-18 under the $L_\infty$ threat model, and the results are summarized in the table below. It is observed that when regularizations are applied only to layer 4, the model can mitigate robust overfitting to a certain extent, but its robustness performance is poor. When regularizations are applied to both layers 3 and 4, the model achieves better adversarial robustness while effectively mitigating robust overfitting. Therefore, in our experiments, we consistently chose to apply regularizations to layers 3 and 4.
> | Method | Layer | Natural(best) | PGD-20(best) | AA(best) | Natural(last) | PGD-20(last) | AA(last) |
> |----------|----------|----------|----------|----------|----------|----------|----------|
> | TRADES-RAT-WP | [-] | 82.77 | 52.67 | 49.28 | 82.94 | 49.65 | 46.80 |
> | TRADES-RAT-WP | [4] | 82.98 | 52.35 | 48.48 | 83.11 | 49.76 | 46.66 |
> | TRADES-RAT-WP | [3,4] | 82.61 | 55.33 | 51.34 | 83.08 | 54.85 | 51.04 |
> | TRADES-RAT-WP | [1,2,3,4] | 82.05 | 55.44 | 51.42 | 82.57 | 54.81 | 50.73 |
>
> >**Q6:** In Figure 4(b), why are the front layers “Layer 1&2&3” while the latter layers are “Layer 3&4”? I am confused with the definition of former vs latter layers here.
>
> **A6:** In this work, the terms “front layers” and “latter layers” are used as relative concepts. Specifically, except for the case of including all layers, when we count from the first layer backward, we regard them as front layers. When we count from the fourth layer forward, we regard them as latter layers. We hope this explanation clarifies the terms used in the paper and addresses your concern.
>
> >**Q7:** Is there any reason for mentioning the semi-supervised learning methods in the Related Work section?
>
> **A7:** Schmidt et al. [1] attributed robust overfitting to sample complexity theory, suggesting that robust generalization requires more training data. This assertion is supported by empirical results in subsequent derivative works. These works leverage semi-supervised learning methods to increase the amount of training data available for adversarial training, which has been shown to effectively alleviate robust overfitting and significantly enhance the adversarial robustness. Considering that these semi-supervised learning methods can help alleviate robust overfitting, we introduced these works in related work.
>
> [1] Schmidt, Ludwig, et al. "Adversarially robust generalization requires more data." Advances in neural information processing systems 31 (2018).
>
> >**Q8:** Where is this evidence for the SVHN dataset, is it in Figure 5?
>
> **A8:** Yes, the evidence can be found in Figure 5(b), which shows the learning curve on the SVHN dataset. As shown in the figure, the model experiences performance degradation before the 25th epoch. However, the learning rate decay of the model occurs at the 100th epoch.
>
> >**Q9:** In Figure 4(c), it is shown that RAT_WP has a negative robust generalization gap throughout most of the training. Can the authors offer some explanations for this unusual phenomenon? What about AWP applied to all layers?
>
> **A9:** This unusual phenomenon is caused by the specific stage at which we calculate the training robust accuracy. For adversarial training that includes weight perturbation, each optimization iteration can be summarized as follows:
>
> 1.	Sampling natural data
> 2.	Generating adversarial data
> 3.	Adding weight perturbation to model parameters
> 4.	Gradient backpropagation and updating model parameters
> 5.	Removing weight perturbation from model parameters
>
> In our experiment, the training robust accuracy of the model is calculated before removing the weight perturbation from the model parameters. This results in a lower training robust accuracy because the model parameters still include the weight perturbation, thus having a negative robust generalization gap. When we calculate the training robust accuracy after removing the weight perturbation from the model parameters, the result is a normal positive robust generalization gap. In AWP, it is the same as above. We hope this explanation provides clarity on the observed phenomenon and addresses your concern.
>
> Thank you once again for your valuable feedback. Please do not hesitate to reach out if there are any further clarifications needed.

---

> > ### Comment · Reviewer_JDM1 · 2024-07-27
> > **Follow-Up by Reviewer JDM1**
> >
> > Thank you for the swift response.
> >
> > A1:This does not fully address my concern. This proof by contradiction relies on an implicit assumption that the contribution to robust overfitting is *linear* in the number of parameters, but that assumption is not verified. We observe an ordinal relationship that latter layers, which have more parameters, contribute more to robust overfitting. It is unclear whether the cause is the greater number of parameters, or there are other factors involved (and if so, what are they?). I still think the paper needs more analysis on why the latter layers mitigate robust overfitting, or, if authors believe so, a more solid argument to convince the audience on why this phenomenon is **not** relevant to the number of parameters.
> >
> > A5: Attaching the results with TRADES is a useful verification. But since TRADES is an AT method, not another network architecture, this result does not fully address my second question on “For other types of network architectures, how does the selection of ‘latter layers’ transfer”. If I understood correctly, the selection of “latter layers” requires numerous experiments with different combinations of layers, under the target {network, attack model and dataset}. Does this mean that to apply RAT_WP under a new experiment setting, we need to conduct these experiments first to select the correct cutoff for “latter layers”?
> >
> > A6: Sorry, my original wording may have led to some misunderstanding. My question was not about “front” vs “latter”. I was trying to ask why, in this Figure, “front layers” are selected differently than in other experiments? As a concrete example, in Section 3.2, authors state that they treat ResNet architecture “as a composition of 4 main layers, corresponding to 4 residual blocks, where the front layers indicate layer 1 & 2 and latter layers indicate layer 3 & 4”. I found this inconsistent with the experiment setup in Figure 4(b) where the front layers are 1 & 2 & 3.
> >
> > A9: Then what is the reason for calculating the robust accuracy before removing weight perturbation?

---

> > > ### Author Response · Authors · 2024-07-27
> > > **Further Response to Reviewer JDM1**
> > >
> > > Thanks for your reply. We will address your remaining concerns below:
> > >
> > > >**Q1:** I still think the paper needs more analysis on why the latter layers mitigate robust overfitting, or, if authors believe so, a more solid argument to convince the audience on why this phenomenon is not relevant to the number of parameters.
> > >
> > > **A1:** There’s a reasonable explanation for why the latter layers help mitigate robust overfitting. For instance, robust overfitting begins in the latter layers and then manifests across the entire network. When we prevent robust overfitting in these latter layers, it effectively stops the entire network from experiencing robust overfitting. However, stopping robust overfitting in the front layers does not have the same preventative effect on the entire network. But it’s important to note that this is just a plausible explanation. Since the mechanisms behind robust overfitting are still unclear, we are afraid we cannot delve deeper into analyzing or explaining it currently due to a lack of supporting evidence. Nevertheless, our claim about the layer-wise property of robust overfitting is well-supported by empirical evidence.
> > >
> > > Sorry, the original response may have led to some misunderstanding. We do not mean that the observed phenomenon is not relevant to the number of parameters. In our revisions, we have also pointed out that the stronger regularization in the latter layers is due to them containing more model parameters. However, this perspective doesn’t seem to fully explain the robust overfitting phenomenon, as the linear relationship between the number of parameters and robust overfitting does not strictly hold in our experiments.
> > >
> > > >**Q2:** Does this mean that to apply RAT_WP under a new experiment setting, we need to conduct these experiments first to select the correct cutoff for “latter layers”?
> > >
> > > **A2:** Yes, in Appendix A.3, we conducted experiments under some popular network architectures. For different architectures, we applied distinct division modes to their network layers. Nevertheless, it is observed that the layer-wise properties of robust overfitting are generally consistent across different network architectures.
> > >
> > > >**Q3:** I found this inconsistent with the experiment setup in Figure 4(b) where the front layers are 1 & 2 & 3.
> > >
> > > **A3:** Thank you for pointing out the inconsistency, and we apologize for the confusion caused. To address your concern, we propose to revise the relevant expressions as follows:
> > >
> > > ```latex
> > > Note that RAT is a general prototype where layer conditions $\mathfrak{C}_{\mathrm{front}}$, $\mathfrak{C}_{\mathrm{latter}}$ and weight adjustment strategy $\mathfrak{S}$ can be versatile. Based on the setting in Section 3.1, the ResNet architecture is treated as a composition of 4 main layers, corresponding to 4 residual blocks. In our subsequent experiments, except for the case of including all layers, when we count from the first layer backward, we regard them as front layers. When we count from the fourth layer forward, we regard them as latter layers. For example, layer 1 \& 2 is $\mathfrak{C}_{\mathrm{front}}$ and layer 3 \& 4 is $\mathfrak{C}_{\mathrm{latter}}$.
> > > ```
> > >
> > > >**Q4:** Then what is the reason for calculating the robust accuracy before removing weight perturbation?
> > >
> > > **A4:** This calculation method is inherited from the official implementation of AWP. Please refer to https://github.com/csdongxian/AWP/blob/main/AT_AWP/train_cifar10.py#L347
> > >
> > > Hope the explanations above could address your concerns. Please let us know if there is more to clarify.

---

> ### Author Response · Authors · 2024-07-22
>
> Dear Reviewer JDM1,
>
> We want to express our appreciation for your valuable feedback, which greatly helped us improve the quality of this paper. If there are any remaining concerns, we would be glad to have further discussions.
>
> Thanks again for your time, and we look forward to hearing from you.
>
> Best regards,
> Authors

---

> ### Author Response · Authors · 2024-07-24
>
> Dear Reviewer JDM1,
>
> We really appreciate your constructive comments that helped us improve this paper. We have made our best effort to address your concerns. We hope our elaborations and the revisions made in the new manuscript could address your concerns. Please let us know if there is more to clarify.
>
> Thanks again for your time, and we look forward to hearing from you.
>
> Best regards,
> Authors

---

> ### Author Response · Authors · 2024-07-26
>
> Dear Reviewer JDM1,
>
> It has been some time since we last heard from you, and we understand that you might be very busy. We just wanted to send a gentle reminder and see if there are any remaining questions or concerns. We sincerely thank you for reviewing our paper and look forward to hearing from you soon.
>
> Best regards,
> Authors

---

### Author Response · Authors · 2024-07-19
**Summary of Revision**

Dear Reviewers,

We sincerely appreciate the time and effort you have dedicated to reviewing our manuscript. We have carefully considered your valuable suggestions and made substantial revisions to our work. The main changes are summarized below:

1. Incorporated the concept of L2 regularization and weight decay in Section 1.
2. Shortened the description of semi-supervised methods in Section 2.
3. Added more analysis of parameter-fixed experiments in Section 3.1, as suggested by Reviewer JDM1 and bT7r.
4. Corrected the experimental results in Table 4, as suggested by Reviewer JDM1.
5. Added comparison between RAT-WP and AWP in Section 4.2, as suggested by all Reviewers.
6. Added ablation study in Section 4.2, as suggested by Reviewer JDM1.
7. Added explanation of the terms “front layers” and “latter layers” in Section 3.2, as suggested by Reviewer JDM1.

Once again, we express our gratitude for your valuable feedback, which have helped us improve the clarity and rigour of our research. Hope our elaborations and the revisions made in the new manuscript could address your concerns. Please let us know if there is more to clarify.

Sincerely,
The authors

---

### Decision · Action_Editor_bk3Z · 2024-09-16

**Recommendation:** Accept with minor revision

**Comment:**

The submission works on analyzing the overfitting issues existing in adverstrail training (e.g., robust overfitting) for addressing adversarial attacks. Previous analyses in such a context do not give a finer analysis per network layer. The proposed research first finds that different layers have varying effects on robust overfitting, and then proposes a robust adversarial training prototype that imposes additional regularization on the latter parts of networks.Two instatiations of RAT verify the efficacy of the proposed research.

Reviewer bT7r tends to think that some of the claims made in the submission are flawed, e.g., "weight decay is not a regularization method that mitigates overfitting", and "it is misleading to claim this phenomena as "a general property of adversarial learning"". While some of these disagreements have been resolved after the rebuttal phase, a few others remain. The key disagreement lies in whether the phenomena identified by the authors are unique to the context of robust overfitting, or they are just general phenomena for training NNs, and the reviewer also expects additional, ideally theoretical, evidence. AE concludes the debates by agreeing that although the identifications and contributions made in the submission are not completely resolving the robust overfitting issues, they show progress along the line and the arguments made in the submission are empirically supported by the experiments, thus satisfying the acceptance criteria of TMLR.

Reviewers CrXW and JDM1 are getting positive on the submission after the rebuttal phase; their decisions are also based on the TMLR acceptance criteria.

For the final version of the paper, please the authors also include experiments for the last few Q%A with the reviewers.

**Audience:**

Yes

**Claims And Evidence:**

Yes

---

> ### Author Response · Authors · 2024-10-05
> **Update Camera Ready Version**
>
> Dear Action Editor bk3Z,
>
> Thank you very much for your appreciation of our work. We have updated the camera-ready version of our paper and included the requested experimental evidence in Appendix A.1.
>
> Best regards,
> The Authors